# Androgen Receptor as an Emerging Feasible Biomarker for Breast Cancer

**DOI:** 10.3390/biom12010072

**Published:** 2022-01-04

**Authors:** Chan-Ping You, Man-Hong Leung, Wai-Chung Tsang, Ui-Soon Khoo, Ho Tsoi

**Affiliations:** Department of Pathology, Li Ka Shing Faculty of Medicine, The University of Hong Kong, Hong Kong SAR, China; u3006037@connect.hku.hk (C.-P.Y.); george09@connect.hku.hk (M.-H.L.); u3006756@connect.hku.hk (W.-C.T.)

**Keywords:** breast cancer, androgen receptor, biomarker, targeted therapy

## Abstract

Biomarkers can be used for diagnosis, prognosis, and prediction in targeted therapy. The estrogen receptor α (ERα) and human epidermal growth factor receptor 2 (HER2) are standard biomarkers used in breast cancer for guiding disease treatment. The androgen receptor (AR), a nuclear hormone receptor, contributes to the development and progression of prostate tumors and other cancers. With increasing evidence to support that AR plays an essential role in breast cancer, AR has been considered a useful biomarker in breast cancer, depending on the context of breast cancer sub-types. The existing survival analyses suggest that AR acts as a tumor suppressor in ER + ve breast cancers, serving as a favorable prognostic marker. However, AR functions as a tumor promoter in ER-ve breast cancers, including HER2 + ve and triple-negative (TNBC) breast cancers, serving as a poor prognostic factor. AR has also been shown to be predictive of the potential of response to adjuvant hormonal therapy in ER + ve breast cancers and to neoadjuvant chemotherapy in TNBC. However, conflicting results do exist due to intrinsic molecular differences between tumors and the scoring method for AR positivity. Applying AR expression status to guide treatment in different breast cancer sub-types has been suggested. In the future, AR will be a feasible biomarker for breast cancer. Clinical trials using AR antagonists in breast cancer are active. Targeting AR alone or other therapeutic agents provides alternatives to existing therapy for breast cancer. Therefore, AR expression will be necessary if AR-targeted treatment is to be used.

## 1. What Are Cancer Biomarkers

The word “biomarker” is derived from the term “biological marker”, referring to a specific indicator of disease in patients that differ from a healthy person, reflecting the connection between a health hazard and a biological state. The well-accepted concept of a biomarker is defined by the US National Cancer Institute (NCI), stating that a biomarker is a biological molecule found in blood, other body fluids, or tissues that is a sign of a normal or abnormal process, condition, or disease. A biomarker can be a protein/peptide, nucleic acid, metabolite, or other categories that may induce a specific clinical symptom. According to the World Health Organization (WHO), it can more broadly be any process that may affect or forecast the occurrence of disease, therapeutic outcomes, disease interventions, and unexpected exposure to environmental factors [1]. Ideally, a biomarker needs to be detected easily, reliably, reproducibly, sensitively, specifically, and cost-efficiently by chemical, physical, or biological assessment. In cancer research, biomarkers in genetic, proteomic, epigenetic, and imaging forms continue to be investigated in various types of cancers. Depending on different clinical applications, cancer biomarkers can be classified into three major types: diagnostic, prognostic, and predictive biomarkers to help narrow down the diagnostic conditions for a specific diagnosis, to provide information regarding the aggressiveness of identified tumors for monitoring disease progression, and estimating the overall outcome of the patient without treatment, and to predict treatment response in order to determine the most effective therapeutic strategy, respectively, each of which provides information for optimizing the clinical care of patients. Some cancer biomarkers serve multiple applications, while some can only satisfy a single purpose [2]. The most frequently used biomarkers in cancers during the past decades were for screening primary and recurrent tumors [3,4]. However, developing novel biomarkers to predict the efficacy of treatment is currently the favored direction. For instance, in breast cancers, the expression status of estrogen receptor α (ERα), progesterone receptor (PR), and human epidermal growth factor receptor 2 (HER2) can be used to guide treatment decisions [5]. This review will mainly focus on discussing the use of androgen receptor (AR) as a prognostic and predictive biomarker for breast cancer management and treatment.

## 2. Cancer Biomarkers Can Guide Therapy

### 2.1. ERα and HER2 as Biomarkers in Breast Cancer Therapy

ERα is one of the estrogen receptors. Unlike estrogen receptor Erβ, which suppresses cell proliferation, ERα functions as a promotor of cell proliferation in breast cancer. Approximately 70% of all breast cancers are found to express ERα (ER + ve breast cancer). The prolonged activation of ERα is known to increase the risk of breast cancer. The receptor is activated upon estrogen stimulation, whereafter it targets downstream signaling pathways. Numerous ERα downstream pathways have been identified, of which the proto-oncogene tyrosine-protein kinase SRC, cyclic adenosine monophosphate (cAMP)/protein kinase A (PKA), mitogen-activated protein kinase (MAPK), phosphatidylinositol 3 kinase (PI3K)/AKT, and phospholipase C (PLC)/protein kinase C α (PKCα) are the most crucial signaling pathways that sustain breast cancer development and progression [6]. Therefore, endocrine therapy, which blocks ERα signaling by either interfering with the binding of ERα and estrogens or by inhibiting the production of estrogens, can effectively suppress tumor growth, thereby improving the prognosis of patients. The selective ERα antagonist tamoxifen and the aromatase inhibitor anastrozole (Arimidex) are commonly used for treating ER + ve breast cancer [7]. For these patients, ERα-targeting therapy is the priority option for disease management. These days, close to 90% of ER + ve breast cancer patients have a survival rate of over five years. This result is attributed mainly to the wide application of ERα-targeting endocrine therapy [8].

HER2 is a transmembrane glycoprotein from the epidermal growth factor (EGF) family. An activating ligand of HER2 has not been identified. It is believed that HER2 is activated in a ligand-independent manner through the formation of homodimers or heterodimers with other EGF members, such as HER1 and HER3, and also with other cell membrane receptors, such as insulin-like growth factor 1 (IGF-1) receptor, which leads to its intracellular domain autophosphorylation [9]. Of all the HER2 dimers, the most activated form has been suggested to be the HER2/HER3 heterodimer [10]. The activation of HER2 stimulates multiple downstream signaling pathways, including the MAPK, PI3K, and PKC. Additionally, activated-HER2 can also down-regulate the activity of p27^kip1^ (p27) protein, a cell cycle suppressor, by the mislocalization of p27 to the cytoplasm and degradation of the protein to enhance the cell cycle [11]. In normal conditions, the protein is expressed in many cells to regulate cell propagation, differentiation, survival, and migration. Around 20% of breast cancers exhibit characteristics of 20-fold overamplification at gene level or up to 100-fold increased expression at the HER2 protein level, which is referred to as HER2-enriched breast cancers [12]. The hyper-activated HER2 has an oncogenic role to drive cancer progression, thus significantly shortening the patients’ disease-free and overall survival. The subsequent successful development of anti-HER2 drugs led to the importance of determining the expression status of HER2 in breast cancers. HER2-targeting drugs, including anti-HER2 antibodies, Trastuzumab, inhibitors, Lapatinib, antibody-drug conjugates such as Enhertu, etc., can recognize and block the activity of HER2, which can be beneficial for patients with HER2 overexpression [13]. Thus, HER2 has become a targetable breast cancer biomarker for instructing clinical therapy.

### 2.2. AR as a Biomarker in Prostate Cancer Therapy and Its Potential Applications in Other Cancers

AR (also known as NR3C4) is a steroid hormone receptor with a molecular weight of 110 kDa. AR belongs to the family of steroid hormone nuclear receptors. The gene encoding for AR is located on the X chromosome (Xq11-Xq12) [14]. Like other typical nuclear receptors, AR contains three major domains and one flexible region involved in the formation of the AR protein. The N-terminal transactivation domain (NTD), the DNA-binding domain (DBD), the hinge region, and the ligand-binding domain (LBD) are located respectively starting from the N-terminal to the C-terminal of the protein (Figure 1a). The NTD contains an activation function 1 (AF1) region responsible for the AR activation process. The DBD is a highly conserved region among the different members of the steroid hormone nuclear receptor family. This domain harbors two zinc finger structural motifs that recognize the target DNA sequence and the AR response element (ARE). A nuclear localization signal (NLS) is located between the DBD and the hinge region. Linked by the hinge region, the LBD is a C-terminal domain; it contains a nuclear export signal (NES) and an activation function 2 (AF2) region. In contrast to the AF1, the activation of AF2 is ligand-dependent. The NES is responsible for the nuclear export of ligand-withdrawn AR, while the NLS is responsible for the nuclear import of activated AR [15]. In the canonical AR signaling pathway, the activation of AR is in a ligand-dependent manner. Without ligand binding, AR is distributed within the cytoplasm, binding to some heat shock proteins (HSPs) to maintain its structural stability and inactivation status. This includes molecular chaperones (HSP90, HSP70, HSP40, HIP, and HOP) and co-chaperones (p23, Cyp40, FKBP51, and FKBP52) [16]. When bound to ligand, AR undergoes a conformational change and dissociates with the HSPs, followed by forming AR homodimers, thus becoming activated. As a transcriptional factor, the activated AR further translocates into the cell nucleus, where it binds to ARE, regulating the expression of AR-targeted genes (Figure 1b). Normal-regulated AR signaling is an essentially biological process for the development and function of the male reproductive system, promoting male sexual differentiation and the induction of male skeletal integrity [17]. In addition, it may also play a role in the regulation of female fertility [18,19]. Abnormal AR signaling contributes to different human diseases, such as androgen insensitivity syndrome (AIS) and prostate cancer [20]. Hence, AR dysregulation has become a sign of certain types of clinical disorders.

The activation of AR signaling will promote cancer development. Prostate cancer is one of the most prevalent malignancies among males, and from existing clinical data, almost all cases are AR-positive tumors [21]. Anti-androgen strategy as prostate cancer therapy can be traced back to the 1940s when Charles Huggins and his colleague found that eliminating androgens can effectively suppress the growth of prostate cancer [22]. Over the past 70 years, the role of AR in prostate cancer have been well studied and characterized. It has come to light that several signaling pathways, including PI3K/AKT/mTOR, RAS/RAF/MEK/ERK1/2, WNT/β-catenin, and non-homologous end joining (NHEJ), are regulated by the activity of AR [23,24,25,26]. AR activation promotes cell cycle, cell metabolism, cell motility, and DNA damage-induced treatment resistance. In prostate cancer, more than 80% of the cases show a response to androgen-blocking therapy. Patients have benefited from androgen deprivation therapy with improved clinical progression, making this treatment a standard treatment for prostate cancer patients. In bladder cancers, the expression of AR has been correlated with tumor progression and poor treatment outcomes [27]. AR signaling could enhance the proliferation and motility of bladder cancer cells in vitro [28,29,30]. In an experimental mouse model, the knockout of AR could offset the incidence of bladder cancer induced by chemical carcinogen [31,32]. In non-small cell lung cancers, the overexpression of AR has been detected in patients, and the knockdown of AR in cell lines could suppress tumorigenicity [33]. Studies have reported that AR is linked to poor prognosis, and AR is involved in several oncogenic signaling pathways to drive hepatocellular carcinogenesis [34,35,36]. In vivo studies have confirmed that the knockout of AR could suppress the occurrence of induced-liver cancers. These studies support the involvement of AR in cancer development [37,38].

## 3. AR as a Biomarker in Breast Cancers

Breast cancer is the most common malignancy in the female population. According to the molecular expression profiles, breast cancers can be classified into five biologically distinct sub-types: luminal A, luminal B, HER2-enriched (HER2 + ve), basal-like, and normal-like [39]. Luminal A and normal-like tumors were characterized by hormone-receptor-positive (ER-positive and/or PR-positive) with HER2-ve and low Ki-67. Luminal B tumors were defined by hormone-receptor-positive with either HER2 + ve or HER2-ve and high Ki-67. The basal-like sub-type lacks ERα, PR, and HER2; it was therefore regarded as triple-negative breast cancer (TNBC). The luminal A sub-type has the best treatment outcome, while the basal-like sub-type has the worst in the clinic [40,41]. The subtyping of breast cancers largely determines the subsequent treatment of the patients. Surprisingly, AR is also prevalent in up to 90 % of all breast cancers [42]. Based on the experience of treating prostate cancer, the possible involvement of AR in the pathogenesis of breast cancer has attracted consideration from investigators. The outcome of clinical studies on AR over the past decades in different sub-types of breast cancers, as documented in Table 1, remain controversial as to whether the AR is a good or poor prognostic factor in breast cancers. Most of the earlier studies were solely focused on the AR molecular profile while ignoring the biological interactions between AR and intrinsic molecular differences in the tumors. Since breast cancers are molecularly heterogeneous, and the growth of the tumor results from the contribution of various molecules, the role of AR in breast cancers needs to be discussed separately for the different sub-types (Figure 2).

### 3.1. The Role of AR in ER + ve Breast Cancer

The expression of AR is often detected in about 60% to 90% of ER + ve breast cancer cases [43,44]. In this sub-type of breast cancer, AR acts as a good prognostic factor. In a study of 931 patients, the survival curves demonstrated that the presence of AR in patients with ER + ve tumors showed a better outcome for disease-free survival (DFS) and overall survival (OS). A study of 1467 postmenopausal breast cancer patients showed similar results [45]. However, the presence of AR would be a poor prognostic factor for ER-ve patients [44]. AR expression in ER + ve/HER2-ve breast cancer was significantly associated with better breast cancer-specific survival (BCS), recurrence-free survival (RFS), and OS; however, AR expression became a poor prognostic factor in ER-ve patients [46]. A study that determined AR’s clinical significance in luminal-B breast cancers showed that the AR + ve cases would have better outcomes for time-to-relapse (TTR) and disease-specific survival (DSS) [47]. Another independent study revealed that high AR expression in ER + ve tumors was associated with less infiltration of lymphocytes, which is a sign of better prognosis, and better survival [48]. Several other studies have also revealed that the expression of AR in ER + ve breast cancer is associated with a smaller size, lower histopathological grading, and lower proliferative properties of the tumors, which might prolong the patients’ survival [44,49,50,51]. These clinical studies supported that AR expression could be a useful prognostic factor in breast cancers.

These findings suggest that AR likely functions as a tumor suppressor in ER + ve breast cancer. This raises the question to investigators: what is the connection between AR and ERα signaling in breast cancer? One of the possibilities is that activated-AR can antagonize the transcription activity of ERα by competitive binding to estrogen responsive elements (EREs). A recently published paper has clarified the detailed mechanism [52]. This study showed that AR activation could replace ERα from chromatin. AR then occupied over 40% of all ERα binding sites (ERBSs), leading to a loss of estrogen response elements (EREs) binding. Meanwhile, ERα was shown to gain new binding targets by relocating to some AR binding sites (ARBSs) to further regulate AR targeted genes, including tumor suppressor *SEC14L2, EAF2,* and *ZBTB16* to inhibit the growth of cells. Furthermore, AR also competes with ERα for binding to a common co-activator, p300, which is essential for the activity of ERα. Since ERα needs a co-regulatory protein SRC-3 to recruit p300 while AR can bind to p300 directly, AR may obtain an advantage in the competition with ERα, resulting in suppression of ER signaling; the activation of AR, therefore, demonstrated an inhibitive effect on ER + ve breast cancer cells [52]. Moreover, AR can inhibit ERα indirectly by some mediator proteins. ERβ is a suppressor of ERα. Activated AR could up-regulate the expression of ERβ gene by binding to the ARE of its promoter region to suppress the activity of ER [53]. In summary, the activation of AR can suppress ER activity by different mechanisms. Since the ERα is a dominant pathway in promoting tumor growth in ER + ve breast cancers, suppressing the ERα can attenuate disease progression. Therefore, AR leads to the better outcome of patients with ER + ve breast cancers.

### 3.2. The Role of AR in HER2 + ve Breast Cancer

Approximately 70% of HER2 + ve/ER-ve breast tumors were detected as AR-positive [46]. In contrast to the ER + ve sub-type, AR + ve patients with a HER2 + ve/ER-ve feature reported a worse clinical outcome in studies. The previous research suggested that AR correlated to the poor DFS and OS in HER2 + ve/ER-ve breast cancer patients [44]. Another study reported that a high mRNA level of AR in HER2 + ve/ER-ve patients was associated with shorter DFS and OS [57]. Studies have demonstrated that AR can crosstalk with HER2 signaling. Such crosstalk could intensify the signaling pathways driven by both AR and HER2 through a positive feedback loop. In the WNT/β-catenin signaling pathway, AR induces the expression of WNT7B to activate the nuclear translocation of β-catenin; AR binds to β-catenin in the nucleus, with the help of FOXA1, leading to the AR/β-catenin complex translocating to the promotor region of *HER3* to promote gene transcription, enhancing the activity of the HER3/HER2 heterodimer [61]. As mentioned earlier, HER2 can activate MAPK signaling [11]. The activated MAPK would induce the expression of AR, which in turn, can enhance HER2 expression. In this loop, AR is essential and adequate for HER2 activation, as AR favors the expression of HER3, while HER2 is crucial for the transduction of MAPK/AR signals [62]. Targeting AR by the shRNA and inhibitor could effectively suppress HER2 + ve/ER-ve breast cancer cell growth in vitro and in vivo [63]. These studies suggested that AR plays an oncogenic role in HER2 + ve breast cancer. 

### 3.3. The Role of AR in TNBC

Around 10% of breast cancer belong to the TNBC sub-group. This sub-type of breast cancer is more aggressive and has a high recurrence risk. The expression of AR was detected in 10–50% of TNBC [42]. In a clinical study, AR + ve TNBC patients were shown to have a decreased survival rate compared with AR-ve TNBC patients [45]. In a study of 559 TNBC cases, the results indicated that AR expression was associated with a worse prognostic outcome in terms of OS; for patients without lymph node metastasis, AR + ve patients had poor OS and DFS, in which the risks of mortality and recurrence were three times higher compared with the AR-ve patients [58]. Similarly, the expression of AR was found commonly in lymph node metastatic TNBC, but rarely in non-lymph node metastatic tumors [64]. Another study showed that AR + ve TNBC patients were more likely to develop a disease recurrence than those unexpressed patients [59]. A study of 263 TNBC patients supported that AR + ve patients would have worse outcomes in five-year distant disease-free survival (DDFS) [60]. Clinical research has associated AR with an inadequate response to neoadjuvant chemotherapy, suggesting the contribution of AR to drug resistance [65]. An in vitro study indicated that AR could promote the survival of TNBC cells, expression of invasion related genes, and thus, metastasis; the inhibition of AR suppressed the metastatic potential of TNBC cells [66]. AR can form a complex with SRC, by recruiting the SRC substrate, focal adhesion kinase (FAK), and the PI3K regulatory subunit, p85α, thus rapidly activating the SRC/PI3K/FAK pathway and its downstream gene, thereby driving cell metastasis [67]. These results suggest that AR can promote the tumor progression of TNBC. In TNBC, activating *PIK3CA* mutations were frequently detected in AR + ve patient samples and cell lines. The PI3K pathway has been revealed to contribute to breast cancer development, while the combined inhibition of AR and PI3K significantly suppressed cell propagation in cell models [68]. These results support that AR can be involved in the pathogenesis of TNBC. The inhibition of AR might suppress progression and reduce the aggressiveness of the disease.

### 3.4. Conflicting Results

Earlier studies highlighted that TNBC patients might benefit from the presence of AR with an improved five-year survival rate, OS, DFS, higher disease-specific survival, and low recurrent risk [69,70,71,72], while the cases with the absence of AR would have a higher risk of tumor metastasis [73,74,75]. A meta-analysis involving 2826 TNBC patients revealed AR expression was related to better DFS and lower tumor grade, but a higher incidence of lymph node metastasis, and no impact on OS [76]. However, another more recent study that analyzed 4914 TNBC patients from 27 studies showed that there was no correlation between AR and patients’ DFS, OS, DDFS, or disease relapse-free survival [77]. The reasons for these contradictory results are still under investigation. Noteworthy, TNBC patients can be further classified into different sub-types by their intrinsic gene profiles. AR + ve luminal TNBCs, known as luminal AR (LAR) sub-type, shows unique characteristics [78]. It has been demonstrated that LAR cancers displayed molecular features similar to luminal A and B breast cancers (ER + ve), including multiple highly reactive hormone-regulated pathways [79]. Interestingly, resembling AR + ve/ER + ve breast cancers, studies have emphasized that patients with LAR type cancers had a favorable prognostic outcome with lower KI-67 levels, lower tumor grade, and higher OS. Moreover, TNBC sub-types were associated with different pathological complete response (pCR) rates to neoadjuvant chemotherapy, with LAR having the worst response, while the basal-like, another TNBC sub-type, had the best response [80]. Furthermore, the differences in correlation between AR with OS among different races and ethnicities has also been reported [81,82]. In around one-third of TNBC cases, the overexpression of ERβ was observed in patient samples, which could suppress the activity of PI3K and AR by upregulating phosphate and tensin homolog (PTEN), further suppressing the cell growth [83]. EGFR and BRCA1 may also affect the function of AR in breast cancers. It has been reported that the EGFR expression level and the frequency of BRCA1 deficiency are higher in TNBC [84]. The co-inhibition of AR and EGFR showed an additive growth suppression [85]. BRCA1 was reported as one of the AR co-activators, while a deficiency in BRCA1 may downregulate the expression of AR, and thus the activity of AR [86]. Therefore, the crosstalk of AR, EGFR, and BRCA1 may affect the significance of AR in breast cancers, especially in TNBC. In prostate cancer, the methylation of CpG islands located in the AR promoter and microRNA modulation leading to the silencing of gene transcriptional activity was reported [87,88]. Whether AR’s expression level and activity in breast cancer are also related to epigenetic modification is poorly understood. A study suggested that 5’ untranslated region mutation (T105A) of AR promotor was identified from AR-negative breast cancer patients, and could affect AR expression [89]. MicroRNAs, for example, miR-34, miR-205, and miR-320, have been reported to modulate the expression of AR in prostate cancer [90]. There should be a similar regulatory mechanism of AR expression in breast cancer. MiR-34 [91] and miR-205 [92] are tumor suppressors in breast cancer. However, the information showing whether these miRNAs would modulate the expression of AR is missing. We do believe some miRNAs would be the upstream regulators of AR expression. Therefore, addressing the upstream regulators of AR will be important in breast cancer. These results may partially explain the conflicting results. In addition, AR-targeted antibodies and the cut-off point for AR positivity (Table 1) used among different studies were diverse. Collectively, these suggest that a more authoritative guidance is needed for determining AR activity in order to help evaluate the clinical significance of AR in TNBC patients.

## 4. Conclusions and Future Perspective

The prognostic values of AR in breast cancers are emerging. In this review, we summarized the latest papers, including clinical studies and nonclinical laboratory research works, to discuss the role of AR in different sub-types of breast cancers. AR functions as a tumor suppressor mainly by antagonizing with ERα in ER + ve breast cancer, and the mechanisms that may be involved in this process have been updated by recently published studies. In contrast, AR performs as a tumor promotor by driving oncogenic pathways in ER-ve breast cancers, including HER2 + ve and TNBC sub-types. More mechanistic studies of AR and more detailed sub-classification of TNBC and standardized protocol in determining AR positivity are required to explain the current conflicting situations, especially in TNBC. 

In some studies, the status of AR in breast cancers has been suggested to be useful as a potential reference for disease management in the future. In AR + ve/ER + ve breast cancer, the AR:ERα ratio has been proposed to predict the treatment outcomes of adjuvant hormone therapy by considering their crosstalk effect on the disease. Patients with a value ≥ 2 showed a higher risk for failure response to tamoxifen. Consequently, the ratio can also predict worse survival of the patients in terms of OS and DSS [93,94]. Additionally, in an analysis of 402 ER + ve patients, tumors with an AR:ERα ratio of ≥2 were expected to be larger in size, with more lymph nodes metastasis, higher proliferative index, higher tumor grade, and risk of tumor recurrence when compared to those with ratio < 2. It was significantly correlated with poor disease-free interval (DFI) and DSS [95,96]. This suggested a threshold of AR levels in ER + ve patients. AR expression exceeding this might offset the molecular function of ERα, resulting in poor response to anti-ERα therapies given to the patients, with the AR pathway possibly governing the disease. Bicalutamide, an anti-androgen drug used in treating metastatic prostate cancer, has been demonstrated to inhibit ER-ve breast cancer cell growth in the pre-clinical study [97]. Notably, several clinical trials targeting AR either alone or combined with other conventional drugs in breast cancer have been carried out or are ongoing (Table 2). For example, there are clinical trials (NCT01889238 and NCT00468715) to determine the efficacy of AR antagonist alone to treat ER-ve breast cancer. A phase II trial (NCT02091960) that combined an AR antagonist with an anti-HER2 antibody, trastuzumab, in advanced AR + ve/HER2 + ve/ER-ve breast cancer groups indicated that around a quarter of the participants exhibit partial response or stable disease during the 24-week treatment period [98]. An ongoing phase II trial (NCT02750358) shows promising results to indicate AR + ve patients’ tolerance to the AR antagonist drug enzalutamide, demonstrating the feasibility of utilizing the anti-androgen drug to treat AR + ve breast cancers [99]. This evidence supports the potential applications of AR antagonists in breast cancer management in the future. The determination of AR status at the time of diagnosis might help the patients benefit from subsequent treatment by deciding an optimal strategy, especially for TNBC, as there is no targeted therapy for this breast cancer sub-type treatment. Thus, in the future, we expect that examining AR will be a route procedure for breast cancer diagnosis, similar to ER.

## Figures and Tables

**Figure 1 biomolecules-12-00072-f001:**
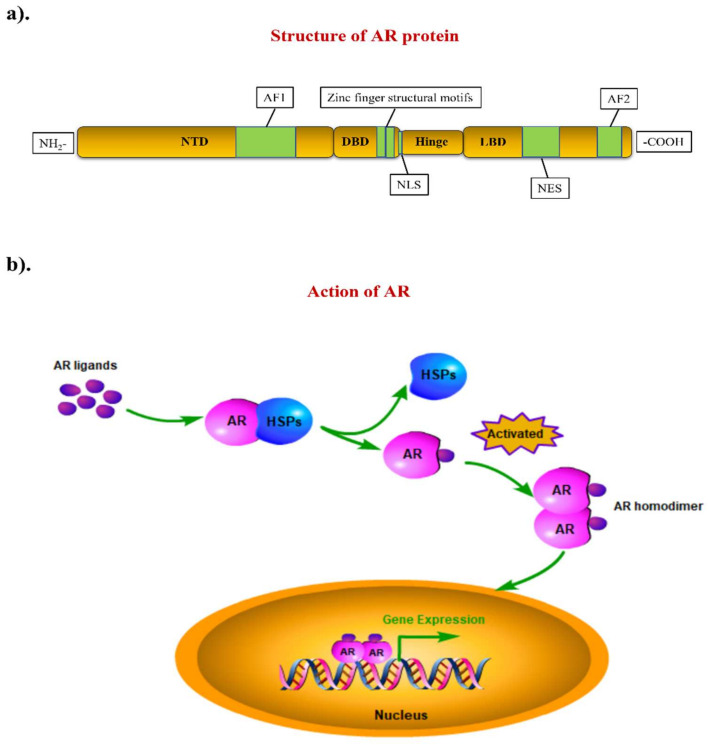
The structure of AR and its molecular mechanism of action. (**a**) The AR gene is located on the X chromosome, it encodes a protein with four main components: the N-terminal transactivation domain (NTD), the DNA-binding domain (DBD), the hinge region and the ligand-binding domain (LBD). (**b**) Unactivated AR is located in the cytoplasm and binds to heat shock proteins (HSPs). When stimulated by androgen, AR separates from HSPs rapidly, followed by forming homodimers and translocating into the nucleus to regulate the expression of its target genes.

**Figure 2 biomolecules-12-00072-f002:**
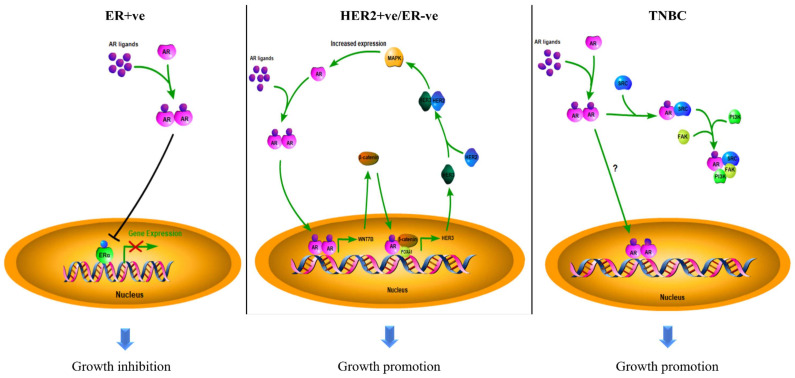
The roles of AR in different sub-types of breast cancer. The mechanisms of action of AR in breast cancers depend on the disease sub-type: AR suppresses ERα in ER + ve cancers to inhibit tumor growth; AR promotes HER2 + ve/ER-ve cell growth by interacting with WNT/β-catenin to induce the expression of HER3, further binding to HER2 to activate the MAPK pathway, which in turn enhances the activity of AR; AR drives TNBC development and progression by activating the SRC/PI3K/FAK pathway. However, the DNA targets of AR are not well characterized in TNBC.

**Table 1 biomolecules-12-00072-t001:** AR in different sub-types of breast cancer has different clinical outcomes.

Types	AR Status (Cut-Off Used to Define AR + ve)	Case No.	Indicator of Clinical Outcomes ^1^	Hazard Ratio (HR)	95% Confidence Interval (CI)	*p*-Value	Reference
ER + ve	Positive (≥10% nuclear-stained)	470	DFS	0.654	0.429–0.997	0.049	[44]
	Negative (<10% nuclear-stained)	202		1	-	-	
	Positive (≥1% nuclear-stained)	1024	OS	0.68	0.52–0.88	-	[45]
	Negative (<1% nuclear-stained)	140		1	-	-	
	Positive (≥1% nuclear-stained)	2833	BCM	0.53	0.41 –0.69	< 0.001	[46]
	Negative (<1% nuclear-stained)	470		1	-	-	
	Positive (≥1% nuclear-stained)	609	DSS	0.259	0.139–0.482	0.000	[47]
	Negative (<1% nuclear-stained)	250		1	-	-	
	High (mRNA Z-score)	145	DRFS	-	-	0.008	[48]
	Low (mRNA Z-score)	144		-	-	-	
	Positive (N/A)	-	DFS	0.40	0.31–0.52	< 0.001	[54]
	Negative (N/A)	-		1	-	-	
	Positive (≥10% nuclear-stained)	909	OS	0.71	0.53–0.95	0.022	[55]
	Negative (<10% nuclear-stained)	162		1	-	-	
	Positive (≥1% nuclear-stained)	461	DFS	0.606	0.388–0.944	0.027	[56]
	Negative (<1% nuclear-stained)	337		1	-	-	
HER2 + ve/ ER-ve	Positive (≥10% nuclear-stained)	49	OS	-	-	0.074	[44]
	Negative (<10% nuclear-stained)	42		-	-	-	
	High (mRNA level)	35	DFS	1.46	1.03–2.06	0.03	[57]
	Low (mRNA level)	49		1	-	-	
TNBC	Positive (≥1% nuclear-stained)	78	OS	1.83	1.11–3.01	0.02	[45]
	Negative (<1% nuclear-stained)	133		1	-	-	
	Positive (≥1% nuclear-stained)	261	OS	2.159	1.224–3.808	0.008	[58]
	Negative (<1% nuclear-stained)	231		1	-	-	
	Positive (≥1% nuclear-stained)	23	DFS	5.26	1.39–19.86	0.014	[59]
	Negative (<1% nuclear-stained)	38		1	-	-	
	Positive (≥1% nuclear-stained)	78	DDFS	1.82	1.10–3.02	0.020	[60]
	Negative (<1% nuclear-stained)	185		1	-	-	

^1^ DFS: Disease free survival; OS: overall survival; BCM: breast cancer-specific mortality; DSS: disease-specific survival; DRFS: distant-relapse-free survival; DDFS: distant-disease-free survival.

**Table 2 biomolecules-12-00072-t002:** Clinical trials of AR-targeted therapies in breast cancers.

ClinicalTrials.gov Identifier (Accessed date: 23 December 2021)	Condition or Disease	Drugs	Phase	Start Date	Status
NCT02091960	AR + ve/HER2 + ve/ER-ve Advanced Breast Cancer	Enzalutamide/Trastuzumab	II	August 2014	Complete
NCT01889238	AR + ve Advanced TNBC	Enzalutamide	II	June 2013	Complete
NCT02457910	AR + ve Metastatic TNBC	Enzalutamide/Taselisib	I/II	June 2015	Complete
NCT00468715	AR + ve/ER-ve/PR-ve Metastatic Breast Cancer	bicalutamide	II	March 2007	Ongoing
NCT03383679	AR + ve TNBC	Darolutamide/Capecitabine	II	March 2018	Ongoing
NCT02750358	AR + ve Early Stage TNBC	Enzalutamide	II	May 2016	Ongoing
NCT02689427	AR + ve Stage I–III TNBC	Enzalutamide/Paclitaxel	II	September 2016	Ongoing
NCT03090165	AR + ve TNBC	Ribociclib/Bicalutamide	I/II	March 2017	Ongoing
NCT02605486	AR + ve Metastatic TNBC	Palbociclib/Bicalutamide	I/II	November 2015	Ongoing

## Data Availability

The datasets generated in this study are available from the corresponding author on reasonable request.

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
