# Peer review of "Androgen Receptor as an Emerging Feasible Biomarker for Breast Cancer"

_biomolecules, 2022, doi:10.3390/biom12010072_

Round 1

Reviewer 1 Report

In this study the Authors aimed to emphasize the importance of androgen receptor as a prognostic and predictive biomarker for breast cancer management and treatment.

It is an interesting topic that could pave the way to novel diagnostic, prognostic and therapeutic approaches for breast cancer patients.

I suggest:

- to deeply discuss the molecular mechanisms involved in the interplay between ERalpha and AR

- to dedicate a separate paragraph describing the possible use of anti-androgen drugs (alone or in combination with other conventional drugs) for breast cancer patients’ management.

Author Response

Reviewer 1

In this study the Authors aimed to emphasize the importance of androgen receptor as a prognostic and predictive biomarker for breast cancer management and treatment. It is an interesting topic that could pave the way to novel diagnostic, prognostic and therapeutic approaches for breast cancer patients.

I suggest:

  1. to deeply discuss the molecular mechanisms involved in the interplay between ERalpha and AR

Response:

Thank you for your suggestions. This is essential information to support AR as a good prognostic factor in ER+ve breast cancer. We have added a paragraph to descript the mechanisms to explain how AR could suppress ERα function. The detailed information is shown in lines 215-225.

  1. to dedicate a separate paragraph describing the possible use of anti-androgen drugs (alone or in combination with other conventional drugs) for breast cancer patients’ management.

Response:

Thank you for your suggestions. We have added a table (Table 2) showing the current active clinical trials involving AR antagonists in breast cancer treatment. These are necessary evidence to demonstrate the usefulness of AR antagonists in breast cancer treatment and management. In addition, knowing the expression status of AR will be important in the future as AR expression can be used to guide the treatment similar to ER. 

Reviewer 2 Report

Overall the paper has come together in a good way. I have some comments as follows.

- What does this paper add to the currently published literature in this area? For example "doi:10.1001/jamaoncol.2016.4975", "PMID: 28861319"

Especially considering that other papers have indicated AR as a biomarker for BC subtypes. For example: DOI: http://dx.doi.org/10.5306/wjco.v6.i6.252

- As this is a review paper, it is expected that relevant papers in the area be included and discussed. For example: https://doi.org/10.1155/2015/357485, https://doi.org/10.1186/1471-2407-12-132

- It is more important to focus on the novel aspects of knowledge and highlight the novelties that this review paper brings. It also should be more highlighted in the abstract.

- It is stated that: "Applying AR expression status to guide treatment in different breast cancer subtypes has been suggested, with promising preliminary results. Thus, AR will in future be a feasible biomarker for breast cancer". How do the "promising results" guarantee certain feasibility of AR as a BC biomarker in future?

- Please be consistent throughout the text in terms of abbreviations. For example, in some subheadings, AR and androgen receptor are used interchangeably. It is suggested to use AR to be consistent. 

Author Response

Overall the paper has come together in a good way. I have some comments as follows.

  1. What does this paper add to the currently published literature in this area? For example "doi:10.1001/jamaoncol.2016.4975", "PMID: 28861319"

Response:

Thank you for your comments which help us shape our review manuscript. We admit that part of the information is overlapped as we cannot exclude the existing facts. We need to include them to provide information for the potential readers who may not be entirely familiar with existing breast cancer research. On top of these previous publications, we have included the most recent studies on AR in breast cancer, including one landscape manuscript, “The androgen receptor is a tumor suppressor in estrogen receptor–positive breast cancer” (Nature Medicine volume 27, pages310–320 (2021)). This provides evidence that AR activation could suppress ER+ve breast cancer. The information in this manuscript can add value to previous clinical studies that demonstrate AR as a good prognostic factor in ER+ve breast cancer. In addition, we have documented the prognosis of AR in different breast cancer sub-types from various clinical studies. By gathering the information in table 1, we can clearly demonstrate that the expression of AR should be a good prognostic factor in ER+ve breast cancer while AR should be a poor prognostic factor in ER-ve breast cancer. This highlights AR functions differently in different situations. In addition, we also include table 2 showing the currently active clinical trials of using AR antagonists in breast cancer therapy. This further demonstrates the importance of AR in breast cancer research.    

  1. Especially considering that other papers have indicated AR as a biomarker for BC subtypes. For example: DOI: http://dx.doi.org/10.5306/wjco.v6.i6.252

Response:

Thank you for your comment. Traditionally, we have been classifying breast cancer based on the expression of ER, PR and HER2. Since 2001, the molecular signature of breast cancer has been proposed. Although 5 molecular subtypes with different prognoses have been suggested, the key features of these subtypes are reflected in ER+ve, ER-ve and HER2 overexpression status. In 2011, Professor Pietenpol’ s group raised a classification system of triple-negative breast cancer (TNBC) by analyzing the genomic profiles of 21 breast cancer datasets (Identification of human triple-negative breast cancer subtypes and preclinical models for selection of targeted therapies J Clin Invest. 2011 Jul;121(7):2750-67. doi: 10.1172/JCI45014.). 7 TNBC sub-types have been proposed. Tumors that express AR and other luminal genes are defined as luminal androgen receptor (LAR) subtype. AR, in some way, has become a biomarker for some breast cancers. As mentioned in our manuscript, LAR is a unique TNBC sub-type that showed similar molecular features to luminal A and B breast cancers, which are ER+ve breast cancer.

This new TNBC classification system is not used in the standard clinical setting. Therefore, we cannot determine the prognostic value of AR in different TNBC sub-types. In our manuscript, we have focused on the role of AR in three major breast cancer subtypes, ER+ve, HER2+ve and TNBC. By reviewing abundant clinical and non-clinical studies, in general, we clarified that AR functions as a tumour suppressor in ER+ve breast cancer while functions as a tumour-promoting factor in ER-ve breast cancer. However, due to the lack of information, we are unable to determine if AR will or will not be a prognostic factor in a particular subtype of TNBC.

  1. As this is a review paper, it is expected that relevant papers in the area be included and discussed. For example: https://doi.org/10.1155/2015/357485, https://doi.org/10.1186/1471-2407-12-132\

Response:

Thank you for your comments. We admit that AR may cross-talk with EGFR and BRCA1, two critical factors in breast cancer development. The cross-talk may modulate the progression of breast cancer. In addition, the mechanisms that govern the expression of AR should be paramount in breast cancer.

We have added information about AR’s epigenetic regulation and miRNA that might regulate AR expression in the revised manuscript. We have added the above content from lines 303-321.

However, a detailed discussion of each of the mechanisms and their interaction is beyond the scope of this manuscript. Therefore, we are sorry that we cannot put comprehensive information related to these issues in our manuscript at the moment. It is a good suggestion to discuss the interaction between AR and these modulators in breast cancer, especially TNBC, and the AR expression level and epigenetic regulation.

  1. It is more important to focus on the novel aspects of knowledge and highlight the novelties that this review paper brings. It also should be more highlighted in the abstract.

Response:

Thank you for your suggestions. We have revised the manuscript to highlight the importance of AR as a prognostic factor that might guide the treatment of breast cancer in the future.

  1. It is stated that: “Applying AR expression status to guide treatment in different breast cancer subtypes has been suggested, with promising preliminary results. Thus, AR will in future be a feasible biomarker for breast cancer and How do the “promising results” guarantee certain feasibility of AR as a BC biomarker in future?

Response:

Thanks for your suggestions. We are sorry we did not make this point clear in the previous manuscript. We have added a table (table 2) and lines 353-364 of the text showing the currently active clinical trials of using AR antagonists for breast cancer treatment. These active trials suggest that AR will be an essential biomarker for the targeted therapy used in breast cancer. The expression of AR in the tumour tissue will be a necessary indicator of the suitability of using AR-targeted therapy. AR-targeted therapy might be the first targeted therapy for TNBC.

  1. Please be consistent throughout the text in terms of abbreviations. For example, in some subheadings, AR and androgen receptor are used interchangeably. It is suggested to use AR to be consistent.

Response:

Thank you for your suggestion. We have corrected it.
